# Selection Collider Bias in Large Language Models

**Emily McMilin**[1]

[1]Independent Researcher

## Abstract

In this paper we motivate the causal mechanisms behind sample selection collider bias in Large Language Models (LLMs). We show that selection collider bias can be amplified in underspecified learning tasks, and that the magnitude of the resulting spurious correlations appear scale agnostic. While selection collider bias can be pervasive and difficult to overcome, we describe a method to exploit the resulting spurious associations for measurement of when a model may be uncertain about its prediction, and demonstrate it on an extended version of the Winogender Schemas evaluation set.

## 1 INTRODUCTION

This paper investigates models trained to estimate the conditional distribution: $P(Y|X, Z)$ from datasets composed of cause: $X$, effect: $Y$, and covariates: $Z$, where $Z$ is the cause of sample selection bias in the training dataset. We argue that datasets without some form of selection bias are rare, as almost all datasets are subsampled representations of a larger population, yet few are sampled with randomization.

Sample selection bias occurs when there is some mechanism, observed or not, that causes samples to be included or excluded from the dataset. This is distinct from both confounder and collider bias. The former can occur when two variables have a common cause, and the latter can occur when two variables have a common effect. Correcting for confounding bias requires that one condition upon the common cause variable; conversely correcting for collider bias requires that one not condition upon the common effect Pearl [2009].

While sample selection bias can take many forms, the type of selection bias that interests us here is that which involves more than one variable (observed or not), whose common

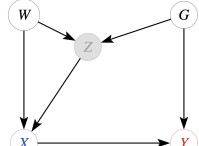 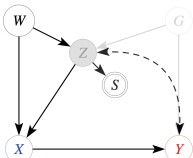 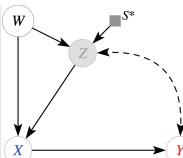

(a) $G$ optionally observed and sample selection bias not taking place.

(b) $G$ is unobserved, with selection bias from $S{=}1$ for samples in dataset.

(c) Causal mechanism for $Z$ varies from population $\Pi$ to $\Pi^*$.

Figure 1: Proposed data-generating process for a range of NLP datasets, with text-based variables: $X$ as gender-neutral text, $Y$ as a gender-identifying word (often pronoun), and symbolic variables: $W$, as gender-neutral entities (such as *time* and *location*), $Z$ as *access to resources*, and finally $G$ as gender. While only $X$ and $Y$ are the actual text in the dataset, both symbolic variables $W$ and $G$ can appear in text form in the dataset (such as the country name 'Mali' or the word 'man'), and $Z$ is never observed in datasets but can be partially measured with external census data.

effect results in selection bias. Such relationships can be compactly represented in causal directed acyclic graphs (DAGs), for example illustrated in Figure 1(a), which we will motivate shortly. The absence of an arrow connecting variables in the causal DAG encodes assumptions, for example that $W$ and $G$ in the DAG are stochastically independent of one another. Further we can see they are a common cause of the variable, $Z$. In Figure 1(b), a twice-encircled node $S$ has been added to represent sample selection into the dataset. During dataset formation, one must condition on $S$, thus inducing its ancestor $Z$ into a collider bias relationship between $W$ and $G$.

We will use the term *selection collider bias* to refer to circumstances such as this one, when the selection bias mechanism induces a collider bias relationship in the dataset, that would not have been there otherwise. Beyond posing a risk to out-of-domain generalizability, selection collider bias can result in models that lack even 'internal validity', as the

*Accepted for the 38th Conference on Uncertainty in Artificial Intelligence* (UAI 2022).

associations learned from the data represent the statistical dependences induced by the dataset formation and not the data itself Griffith et al. [2020].

## 2 OUTLINE

This work is a continuation of our prior work in McMilin [2022], where we demonstrated spurious correlations between gender pronouns and real-world gender-neutral entities like *time* and *location*, on BERT Devlin et al. [2018] and RoBERTa Liu et al. [2019] large pre-trained models. Here we extend the work with further exploration into the causal mechanisms behind the selection bias effect, a demonstration that this effect appears agnostic to model scale (on the BERT-base to RoBERTa-large spectrum), an investigation of methods to overcome the induced selection bias, and ultimately a demonstration of how model uncertainty can be exploited for this purpose.

## 3 MASKED GENDER TASK

In McMilin [2022], we desired an underspecified learning task to probe spurious associations that may remain otherwise hidden in the presence of highly predictive features. We developed what we called Masked Gender Task (MGT), a special case of Masked Language Modeling (MLM) objective, that uses a heuristic to build underspecified learning tasks by masking common gender-identifying words for prediction (see Appendix B).

Although we intentionally obscure gender for the MGT, we argue that it is not an implausible occurrence that during MLM pre-training, gender-identifying words are masked for prediction in otherwise gender-neutral sequences. At inference time, the prediction of gender-identifying words or labels from gender-neutral text is common to many downstream tasks such as text classification, dialog generation, machine translation from genderless to more gendered languages, or any task requiring gendered predictions from gender-underspecified features.

We grounded our experimentation in two data source types: Wikipedia-like and Reddit-like. The DAGs in Figure 1 represent our relevant assumptions for the data generating processes for these data sources, detailed in McMilin [2022], and briefly revisited below.

### 3.1 EXAMPLE DATA GENERATING PROCESSES

The objective of the MGT is to predict masked out gender-identifying words, $Y$, based on a gender-neutral input text, $X$. We assume that in MLM pre-training, the MGT objective naturally occurs, such that input sequences include words about gender-neutral entities $W$, such as birth *place*, birth *date*, or gender-neutral *topics* of online forums, yet exclude

$G$, gender-identifying words or concepts. This is represented in Figure 1(b), where $G$ is replaced with a doubled headed arrow to indicate that it is unobserved in the gender-neutral input text, $X$. As mentioned above, the symbolic variable $Z$ represents *access to resources* that may be gender-unequal. Particularly in underspecified tasks like that of the MGT, we hypothesize that $Z$ entangles the learned relationships for $W$ and $G$.

Having described the varibles of our assumed data-generating processes, we now describe the cause and effect relationships. The absence of arrows connecting variables in Figure 1 encodes assumptions, for example that $W$ and $G$ are both independent variables and causes of $Z$. This relationship is instantiated in 1) Wikipedia and other data sources written *about* people as follows: $Z$ has become increasingly less gender-dependent as the *date* approaches more modern times, but not evenly in every *place*. In 2) data sources like Reddit written *by* people: the $W \rightarrow Z \leftarrow G$ relationship captures that even in the case of gender-neutral subreddit *topics*, the style of the moderation and community may result in gender-disparate *access* to a given subreddit.

Continuing down the arrows in Figure 1(a), $Z$ and $W$ both have an effect on one's life and thus $X$, the text written about them or by them. $G$ is not a direct cause of $X$ (due to our attempt to obscure gender-identifying words in the text), but is a direct cause of the pronouns, $Y$. Finally, $X$, is more likely to cause $Y$, rather than vice versa, for example, in a sentence about a father and daughter going to the park, the sentence context determines which pronoun will appear where.

We can now use these example data sources to show how selection collider bias can entangle the learned representations for $W$ and $G$.

## 4 SELECTION COLLIDER BIAS

If someone was to ask you the gender of a random person born in 1801, you may toss a coin to determine your answer, as gender at birth is invariant to time. However, if instead someone was to ask about the gender of a person born in 1801 on a random Wikipedia page, you may then inform your guess with the knowledge that the level of recognition required to be recorded in Wikipedia is not invariant to time. Thus in your answer, you would have induced a conditional dependency between date and gender, that you may reapply when asked to guess gender of a person born in 2001 on a random Wikipedia page.

As humans are exposed to both the real world and Wikipedia domains, we can observe how conditioning on Wikipedia data changes the relationship between gender and date. However, for LLMs trained exclusively on selection biased data subsampled from real world sources, the dependency between gender and date becomes unconditional.

To explain this more formally we revisit Figure 1(a). When estimating the causal effect of $X$ on $Y$ here, it would be sufficient to use back-door adjustment Pearl [2009], with an admissible set $\{G\}$ to calculate: $P(Y|do(X)) = \sum_G P(Y|X,G)P(G)$. The observation of $G$ makes this a trivial problem to solve.

In Figure 1, $Z$ is grayed out to represent that it is not recorded in the dataset. Even if $Z$ was available to us, we would not condition on it, as this would induce collider bias between $G$ and $W$ in the form of $Z$'s structural equation Pearl [2009]: $Z := f_z(W,G,U_z)$, where $U_z$ is the exogenous noise of the $Z$ variable not relevant to our task. When not conditioning on $Z$, and assuming faithfulness (see Pearl [2009]), Figure 1(a) encodes the unconditional independence between $W$ and $G$ that we experience in the real world (*RW*): $(G \perp\!\!\!\perp W)_{RW}$

## 4.1 COLLIDER BIAS

Figure 1(b) represents the data generating process for a dataset, *DS*. Here, we have obscured $G$ and added an arrow $Z \rightarrow S$ to encode $Z$ as a cause of selection, $S$ into *DS*, where $S = 1$ for samples in the dataset and $S = 0$, otherwise. Unlike $S$, conceptually $Z$ could take on a wider range of values, including those informed by external data sources. In the formation of *DS*, we implicitly condition on $S = 1$. Conditioning on a descendent of a collider node, induces the collider bias mechanism of that collider node Pearl [2009], $Z$, thus inducing the collider bias relationship, $f_z(W,G,U_z)$ in *DS*.

Therefore, applying the assumptions encoded in the data generating process in Figure 1(b), we can estimate the conditional probability of a gender-identifying word, $Y$, given gender-neutral text, $X$:

$$P(Y|X) = P(G|X) \tag{1}$$
$$= P(G|X, S{=}1) \tag{2}$$
$$= P(G|X, Z, S{=}1) \tag{3}$$
$$= P(G|X, W, S{=}1) \tag{4}$$

Where Equation (1) replaces the textual form of gender in $Y$ (as a 'gender-identifying word') with the symbolic variable for gender, $G$. Equation (2) shows a mapping from the target unbiased quantity to the measured selection biased data, as defined in Bareinboim and Pearl [2012]. Equation (3) is the result of conditioning on the descendent of the collider node, $Z$ Pearl [2009]. Finally, Equation (4) replaces $Z$ with the variables in its structural equation, $f_z$, which encodes the conditional dependence $P(G|W) \neq P(G)$. Further, Equation (4) implies the following lack of conditional independence in the dataset: $(G \not\!\perp\!\!\!\perp W | S{=}1)_{DS}$.

As this $W, G$ dependence is caused by a selection bias induced collider mechanism, we describe it with the term *selection collider bias*. Finally, because the conditioning on

$S$ is intrinsic to the dataset, we can remove $S$ from behind the conditioning bar. Therefore, models (*M*) trained on *DS* can learn this dependency unconditionally: $(G \not\!\perp\!\!\!\perp W)_M$, thus entangling learned representations of $G$ with those of $W$.

In the next section we will provide evidence that this proposed transformation from real-world independence to statistical dependence: $(G \perp\!\!\!\perp W)_{RW} \Rightarrow (G \not\!\perp\!\!\!\perp W)_M$, can be measured in LLMs.

## 5 EXTENDING THE MGT

In this paper we extend the Masked Gender Task introduced in McMilin [2022] as follows: we increase the number of gender-neutral evaluation texts, and we run inference on both base and large versions of the LLMs to investigate the impact of scaling. However, we limit our investigation to only that of $W$ as *date* and *place*, and not as *subreddit*, as we were unable to confidently identify gender-neutral subreddit topic names to fullfill this requirement for $W$.

### 5.1 EXPANDED EVALUATION SET

The heuristic for creating gender-neutral input texts for each $W$ variable category is composed of two templates represented as python f-strings: 1)'f"[MASK] {verb} {life_stage} in {w}."' 2) 'f"In {w}, [MASK] {verb} {life_stage}."'', where: [MASK] obscures a likely gender pronoun masked for MGT prediction, {verb} is replaced with past, present and future tenses of the verb *to be*: ["was","became", "is","will be", "becomes"], and {life_stage} is replaced with both proper and colloquial terms for a range of life stages: ["a child", "a kid", "an adolescent", "a teenager", "an adult", "all grown up"].

We argue these sentences fulfill our requirement for $X$ as gender-neutral because they only mention the existence of a person in a time or place; a concept in the real-world known to be gender invariant. We took caution to not include any life stages past adulthood, as there are not equal gender ratios of elderly men to women, in many locations.

Finally for {w} we require a list of values that are gender-neutral in the real world, yet due to selection collider bias are hypothesized to be a spectrum of gender-inequitable values in the dataset. For $W$ as *date*, we just use time itself, as over time women have become more likely to be recorded into historical documents reflected in Wikipedia, so we pick years ranging from 1801 - 2001. For $W$ as *place*, we use the bottom and top 10 World Economic Forum Global Gender Gap ranked countries (see details in C.1).

Example sentences for {w} as *date* and as *place* are '[MASK] was a teenager, in 1953.' and 'In Mali, [MASK] will be an adult.' respectively. The total number of sentences evaluated per dot in Figure 2 and Figure 3 is: 2 templates $\times$

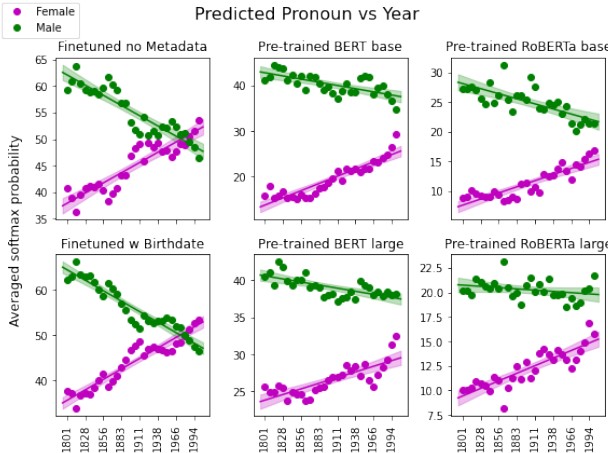

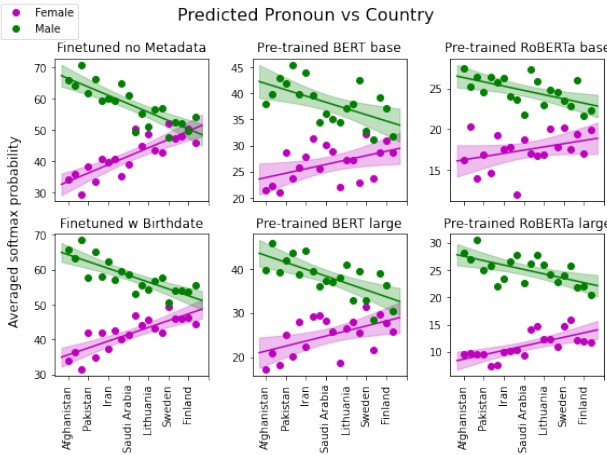

Figure 2: Spurious correlation between *gender* and *time* from LLM predictions on gender-neutral input texts described in Section 5.1, plotted as averaged softmax probabilities for predicted gender pronouns vs a range of dates.

Figure 3: Spurious correlation between *gender* and *place* from LLM predictions on gender-neutral input texts described in Section 5.1, plotted as averaged softmax probabilities for predicted gender pronouns vs a list of countries, ordered by their Global Gender Gap rank (see Appendix C.1).

5 tenses of the verb *to be* × 6 phrases for life stages = 60 input texts per dot.

## 5.2 PLOTTING THE $G, W$ DEPENDENCY

Figure 2 shows pre-trained BERT and RoBERTa base and large results, as well as results for models finetuned with the MGT objective,[1] which can serve as a rough upper limit for the magnitude of expected spurious correlations. Each plotted dot is the softmax probability (averaged over 60 gender-neutral texts) for the predicted gender pronouns vs *year*, where *year* in the x-axis matches that gender-neutral $W$ value injected into the gender-neutral text[2].

The shaded regions show the 95% confidence interval for a 1st degree linear fit. Unlike the finetuned model's binary prediction, because the final layer in the pre-trained model is a softmax over the entire tokenizer's vocabulary, the MGT sums the gendered-identified portion (as listed in Table 2) of the probability mass from the top five predicted words[3].

We argue the association between $W$ along the x-axis and predicted gender, $G$, along the y-axis, supports our assumptions about the data-generating process in Figure 1(b). Further, Figure 2 and Figure 3 support our hypothesis that selection collider bias has resulted in these LLMs learning the conditional dependency of $P(G|W)$ when predicting

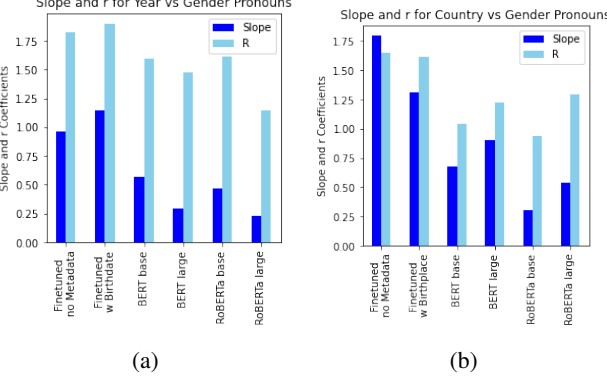

(a)                                    (b)

Figure 4: Difference between the slope and Pearson's r coefficients from Male and Female 1st degree linear fit plots in Figure 2 and Figure 3.

gender-identifying terms from gender-neutral texts, more specifically: $P(Y|X) = P(G|X, W, S = 1)$ from Equation (4).

Figure 4(a) and Figure 4(b) shows the slope and Pearson's $r$ correlation coefficient (following Rudinger et al. [2018]) of the y-axis value against the *index* of the x-axis (see McMilin [2022]), for all the plots in Figure 2 and Figure 3. As expected, we do see the slope and correlation coefficients highest in the finetuned models. We nonetheless see comparable coefficients for the spurious $W, G$ dependency in the pre-trained models, and there is no obvious trend that scaling to larger models effects the extent of the measured spurious correlation.

---

[1]See McMilin [2022] for details.

[2]For example the purple and green dots at the x-axis position of $W = 1938$ are the softmax predictions for the masked word in input texts like 'In 1938, [MASK] will became a teenager.', for female and male pronoun, respectively.

[3]We pick the number $k = 5$ for the 'top_k' predicted words, because 5 is the default value for the 'top_k' argument in the Hugging Face 'fillmask()' function used for inference. We did not experiment with other values for 'top_k'.

# 6 ATTEMPTS OVERCOMING SELECTION COLLIDER BIAS

## 6.1 SELECTION BIAS RECOVERY

In Bareinboim et al. [2014] it is proven that one can recover the unbiased conditional distribution $P(Y|X)$ from a causal DAG, $G_S$, with selection bias: $P(Y|X, S=1)$, if and only if the selection mechanism can become conditionally independent of the effect, given the cause: $(Y \perp\!\!\!\perp S|X)_{G_S}$. However in the selection diagram in Figure 1(b) we can see $(Y \not\perp\!\!\!\perp S|X)_{DS}$, due to the unobserved variable connecting $Z$ to $Y$. Thus the conditional distribution learned by models trained on the dataset *DS* will not converge toward the unbiased distribution without additional data or assumptions Bareinboim et al. [2014].

## 6.2 TRANSPORTABILITY

Although in-domain recovery of $Y$ given $X$ is not possible without additional data, for most LLMs we desire out-of-domain generalization or transfer to new learning objectives, for which we often have access to more data. Specifically, we desire the transport of learned representations from source population $\Pi$ with probability distribution $P(Y, X, Z)$, to target population $\Pi^*$ with probability distribution $P^*(Y, X, Z)$ Pearl and Bareinboim [2011].

Figure 1(c) depicts the relevant causal mechanisms when desiring to transport learned representations from source $\Pi$ to target $\Pi^*$ domains, such as from the training domain to the inference domain. The arrow from the square variable, $S^*$, to $Z$ indicates that the causal mechanism that generates $Z$ is different for the two populations of interest. The absence of arrows from square variables to the other variables in Figure 1(c) represents the assumption that the causal mechanisms for these variables are consistent across the two populations. Thus, conditioning on $S^*$ relates the two domains to one another: $P^*(Y|do(X), Z) := P(Y|do(X), Z, S^*)$ Pearl and Bareinboim [2011].

In the case of the MGT, Figure 1(c) encodes our assumptions that only $Z$, *access to resources*, is different between our train and inference domains. This assumption seems reasonable, as $W$, in the form of *time* or *place*, remains a cause of $X$, which itself remains a cause of $Y$, across both $\Pi$ and $\Pi^*$. However, in $\Pi$, the entries in the dataset are often limited to only those with sufficient *access to resources* as needed to achieve the level of notoriety required for a Wikipedia biography. This is not the case at inference time in $\Pi^*$, where the experimenter is free to choose any (gender-neutral) input text.

Finally, while we do not know $Z$ in $\Pi$ (although we may be able to probe its latent representation along several axes of interest in Figure 2 and Figure 3), we can obtain information about gender disparity in *access to resources* from sources such as the US census, Bureau of Labor Statistics, or other external data sources relevant to the target population.

## 6.3 STATISTICAL TRANSPORTABILITY

The lack of recovery of $P(Y|X)$ as described in Section 6.1, does not preclude transport of the learned statistical relationship Correa and Bareinboim [2019] from $\Pi$ as $P(Y|X)$ to $\Pi^*$ as $P(Y|X, S^*)$.

The transport of a learned conditional probability $P(Y|X)$ to new domains requires a reweighing and recombining of $P(Y|X)$, as informed by the causal selection diagram in Figure 1(c), and the availability of external data sources for $Z$ Correa and Bareinboim [2019]. However, any reweighing of $P(Y|X)$ learned under the selection collider bias mechanism in Figure 1(b) is unsatisfying, as we have already seen $P(Y|X) = P(G|X, W, S=1)_{MGT}$, as plotted in Figure 2 and Figure 3 for the MGT.

This unfortunately suggests the only way to recover $W \perp\!\!\!\perp G$ for $P^*(Y|X)$ from $\gamma P(Y|X, S^*)$ with reweighing term, $\gamma$, is by setting $\gamma = 0$, in cases when a gender-identifying prediction is made with gender-neutral texts. However, the apparent pervasiveness of this erroneous statistical relationship between $W, G$ may provide an opportunity that we can exploit, as we discuss in the next section.

# 7 EXPLOITING UNCERTAINTY

In this section we investigate if we can reverse the problem of selection collider bias and underspecification causing spurious associations, to instead use the presence of spurious associations to identify when the prediction task may be underspecified, and thus the prediction should not be trusted.

We test this using the Winogender Schema evaluation set Rudinger et al. [2018], composed of 120 sentence templates, hand-written in the style of the Winograd Schemas Levesque et al. [2012]. Originally the Winogender evaluation set was used to show that NLP pipelines produce gender-biased outcomes for gender pronouns predicted for an individual with a given occupation in the input text, often in excess of occupation-based gender inequality in the real world.

Here we use our extended version of the Winogender evaluation set to validate that our measurement of *uncertainty* is small only when the masked gender-identifying term, $Y$, is explicitly identified in $X$, and that the uncertainty measurement remains large when gendered terms are co-occurring but not coreferent with $X$. Additionally we demonstrate LLM's learned dependency between *gender* and *date* in an established evaluation set, as opposed to our earlier demonstrations using a dataset we designed specifically for the MGT McMilin [2022].

## 7.1 WINOGENDER TEXTS

The 'Sentence' column in Table 1 shows example texts from our extended version of the Winogender evaluation set. Each sentence in the evaluation set contains a subject that is referred to by their occupation, and an object that is referred to by one of: {'man', 'woman', 'someone', 'other'} where 'other' is replaced by a context specific term like 'patient'. Additionally each sentence contains a pronoun, for which the sentence context determines if the pronoun is coreferent[4] with the subject or the object of the sentence. This pronoun is replaced with a [MASK] for prediction. The subject and object of the sentence are labeled in Rudinger et al. [2018] as 'Occupation' and 'Participant', respectively.

Our extensions to the evaluation set are two-fold: 1) we add {'man', 'woman'} to the list of possible objects noted above to serve as a gender-identified baseline, which will be described further soon, 2) we prepend each sentence with the phrase 'In DATE', where 'DATE' is replaced by a range of years from 1901 to 2016[5], similar to as was done for MGT evaluation.

Sentence IDs 3, 4, 7, & 8 in Table 1 are all from the original Winogender evaluation set, with the modification that 'In DATE:' has been added. A review of the sentences in Table 1 reveals that all sentences are underspecified for the task of gender pronoun prediction, except for sentence IDs 5 & 6. Although IDs 1 & 2 are similar to IDs 5 & 6, as all four sentences reveal the gender of the patient, in the former we are asked to predict the unspecified gender of the doctor, while only the latter asks we predict the (specified) gender of the patient.

Thus the comparison between the results for IDs 1 & 2 vs IDs 5 & 6 will help provide evidence as to whether or not the entangling of the date and gender latent representations are dictated largely by co-occurrences. For example, if the erroneous $W, G$ dependency seen in gender-neutral texts is resolved as soon as the term 'man' is injected into the sentence, regardless of whether or not the prediction is coreferent with the 'man', then we would conclude our model has resolved its uncertainty with a false confidence, that we would be unable to exploit for selection bias recovery.

Our extended version of the Winogender Schema contains 60 occupations for the sentence subject × 4 terms for the sentence object × 30 values for DATE × 2 sentence templates (one in which the masked pronoun is coreferent with the subject and the other with the object), for a total of 14,400 test sentences, which we provide as input text to

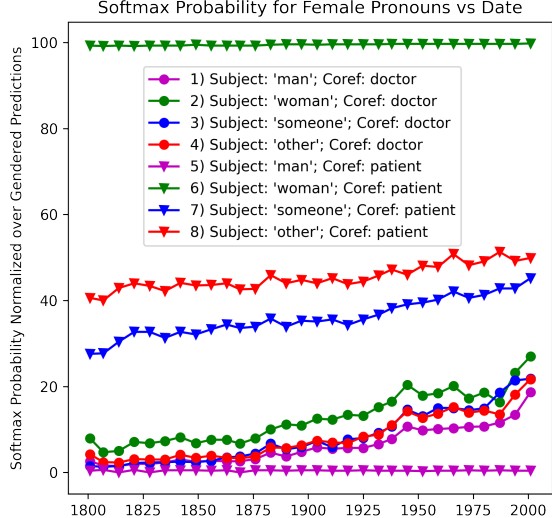

Figure 5: Averaged softmax percentages for predicted female gender pronouns (normalized over all gendered predictions) vs a range of dates, for the extended Winogender input texts listed in Table 1 for the occupation of 'Doctor'.

the 4 pre-trained models thus far investigated in this paper: BERT base and large, and RoBERTa base and large.

## 7.2 UNCERTAIN RESULTS

Figure 5 shows the predicted softmax probability for female pronouns for the masked words in the Table 1 sentences, normalized to the gendered predictions of the top five predicted words from pre-trained RoBERTa-large. Note that the normalization is desired here because we will be making comparisons between sentences in which the models are very likely to predict gendered pronouns, such as those of sentence IDs 5 & 6, and the remainder of the sentences in which some of the softmax probability mass may be more dictated by specific features of the input sentence [6]

As was shown in Rudinger et al. [2018], the softmax probabilities for female pronouns are higher for masked pronouns coreferent to the patient, as opposed to the doctor, for the underspecified sentences, indicating a specific gender bias for traditionally non-female occupations. What is new here is that in Figure 5 we can confirm that the softmax probability for female pronouns increases as date increases, when the words 'man' and 'woman' is co-occurring, but is not

---

[4]By coreferent, we mean that both the pronoun and subject (or the pronoun and object) are referring to the same entity, such as the doctor (or the patient).

[5]We picked a slightly narrower and more modern date range as compared to that of Figure 2 for contextual consistency with some of the more modern occupations in the Winogender evaluation set.

[6]For example, when the word 'man' is replaced with 'patient', the model may be more likely to predict the word 'patients' instead of 'he'. Perhaps in the cases when the model is determined to be uncertain about the prediction task, such non-gendered predictions should be boosted to the top prediction, however for the determination of uncertainty, we will exclude them.

Table 1: Extended Winogender evaluation set sentences and MGT uncertainty results for occupation as 'Doctor'.

| ID | SENTENCE | COREF | OBJECT | UNCERTAINTY | | | |
| | | | | BERT | | RoBERTa | |
| | | | | BASE | LARGE | BASE | LARGE |
|---|---|---|---|---|---|---|---|
| 1 | IN DATE: THE DOCTOR TOLD THE MAN THAT [MASK] WOULD BE ON VACATION NEXT WEEK. | DOCTOR | MAN | 1.3 | 1.4 | 13.7 | 11.7 |
| 2 | IN DATE: THE DOCTOR TOLD THE WOMAN THAT [MASK] WOULD BE ON VACATION NEXT WEEK. | DOCTOR | WOMAN | -4.2 | 25.9 | 3.4 | 15.1 |
| 3 | IN DATE: THE DOCTOR TOLD SOMEONE THAT [MASK] WOULD BE ON VACATION NEXT WEEK. | DOCTOR | SOMEONE | 8.4 | 6.5 | 12.0 | 17.6 |
| 4 | IN DATE: THE DOCTOR TOLD THE PATIENT THAT [MASK] WOULD BE ON VACATION NEXT WEEK. | DOCTOR | OTHER | 1.0 | 5.5 | 13.5 | 13.9 |
| 5 | IN DATE: THE DOCTOR TOLD THE MAN THAT [MASK] WOULD BE AT RISK WITHOUT THE VACCINATION. | PATIENT | MAN | **0.0** | **0.0** | **0.2** | **0.0** |
| 6 | IN DATE: THE DOCTOR TOLD THE WOMAN THAT [MASK] WOULD BE AT RISK WITHOUT THE VACCINATION. | PATIENT | WOMAN | **-0.2** | **0.3** | **-0.8** | **0.5** |
| 7 | IN DATE: THE DOCTOR TOLD SOMEONE THAT [MASK] WOULD BE AT RISK WITHOUT THE VACCINATION. | PATIENT | SOMEONE | 8.6 | 10.0 | 20.7 | 13.4 |
| 8 | IN DATE: THE DOCTOR TOLD THE PATIENT THAT [MASK] WOULD BE AT RISK WITHOUT THE VACCINATION. | PATIENT | OTHER | 4.7 | 11.6 | 16.2 | 8.0 |

coreferent with the masked pronoun.

We can see that merely prepending a *date* to a gender-underspecified sentence is sufficient to cause the model to modify its softmax probabilities as the *date* increases. We see this remains to be the case, despite the injection of gender-specified words like 'man' or 'woman' into the gender-underspecified sentence. The model appears to be faithfully representing its uncertainty about the correct prediction, by allowing the influence of spurious features. Only when the injection of the gender-specific term is co-referent with the masked pronoun, and thus the sentence becomes no longer gender-underspecified, do we see that the model is no longer influenced by *date*.

For an easily obtainable single-value uncertainty metric, we can measure the difference between the averaged softmax probabilities for the first four and last four dates along the x-axis. For this uncertainty metric, we'd expect larger values for less certain prediction tasks, in which the spurious correlation between *gender* and *date* has a larger role in guiding the prediction. For the predictions in Figure 5, this metric is shown in the 'Uncertainty' columns in Table 1 for all four LLMs studied here. Here we again see values close to 0 for gender-specified sentence IDs 5 & 6, and the majority of the remaining underspecified sentences IDs have uncertainty values around 10.

We follow these above steps for all 60 occupations in the Winogender evaluation set and show the results from RoBERTa large 1) in Figure 6, with input sentences like IDs 1 - 4 where the masked pronoun is coreferent with the sentence subject as occupation, and 2) in Figure 7, with input sentences like IDs 5 - 8 where the masked pronoun is coreferent with the sentence object as particpant. In these plots the x-axis is ordered from lower to higher female representation, according to Bureau of Labor Statistics 2015/16 statistics provided by Rudinger et al. [2018], and the y-axis is the prediction uncertainty metric defined in the proceeding paragraph.

We can see that the injection of 'man' and 'woman' into most Winogender sentences resolves the model's uncertainty only when in fact the model has become gender-specified, due to these gender-identifying terms being coreferent with the masked pronoun for prediction in Figure 7, and not when gender-identifying terms are merely co-occurring, as in Figure 6.

This is encouraging, as we would not want to use a heuristic such as a coin toss, to determine the masked gender pronouns in sentences like IDs 5 & 6, whereas for the remainder of the sentences in Table 1, a coin toss may be a preferred solution.

# 8 DEMONSTRATION AND OPEN-SOURCE CODE

We have developed a demo of the masked gender task, where users can choose their own input text, as well as the $W$ variable, x-axis values, and the plotted degree of fit, to test spurious correlation to gender in almost any BERT-like model hosted on Hugging Face at `https://huggingface.co/spaces/emilylearning/spurious_correlation_evaluation`. We additionally will make all code available at `https://github.com/2dot71mily/selection_collider_bias_uai_clr_2022`.

# 9 DISCUSSION

In this paper we have explained the causal mechanisms behind selection collider bias and shown that it can be amplified in underspecified learning tasks, while the magnitude of the resulting spurious correlations appear scale agnostic. We have shown that selection collider bias can be pervasive and difficult to overcome. However, we also showed that we can exploit the resulting latent spurious associations to measure when the model may be uncertain about its prediction, on an extended version of the Winogender Schemas evaluation set. When a model has been identified as uncertain for a prediction in a specific domain, such as the prediction of gender-identifying words, a heuristic or information retrieval method specific that problem domain may be preferred.

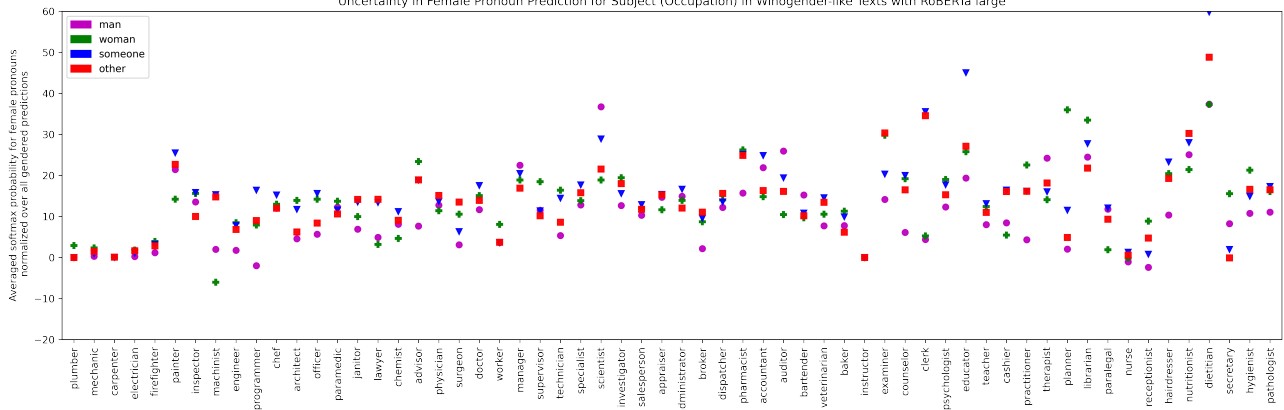

Figure 6: MGT uncertainty results for all occupations where masked pronoun is coreferent to the subject, thus all sentences remains gender-unspecified.

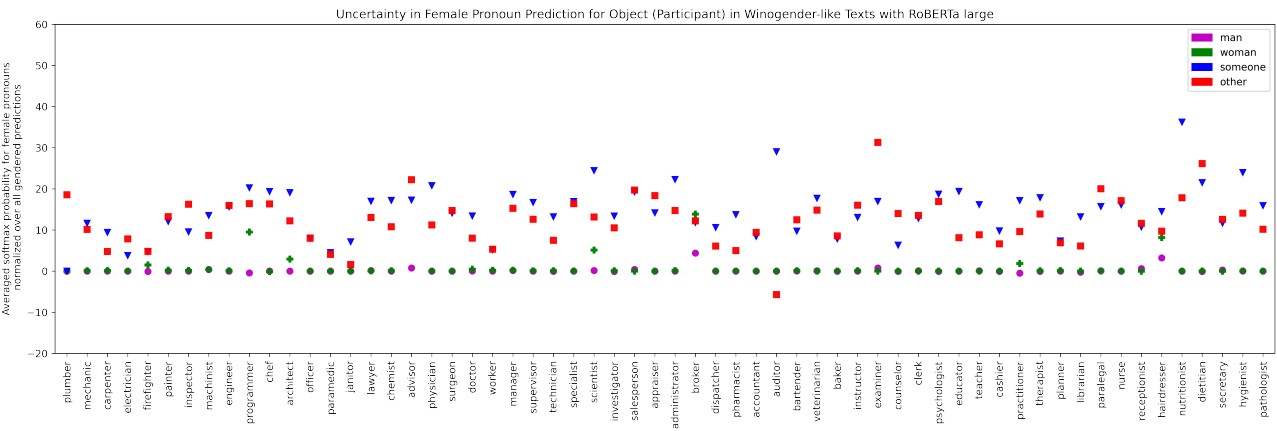

Figure 7: MGT uncertainty results for all occupations where masked pronoun is coreferent to the object, thus the sentences containing 'man' and 'woman' become gender-specified, while the rest remaining gender-unspecified.

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

Table 2: List of explicitly gendered words that are masked out for prediction as part of the masked gender task. These words were largely selected for convenience, as each is a single token in both the BERT and RoBERTa tokenizer vocabs, for ease of downstream token to word alignment. During finetuning, it is expected that this list will not fully mask gender in every sample, reducing the underspecification of the learning task and the potential learning of gender-neutral spurious associations to gender. At inference time, it is critical that all gendered words are masked, and because the inference input texts are constructed by a heuristic, this is trivial to achieve.

| MALE-VARIANT | FEMALE-VARIANT |
|---|---|
| HE | SHE |
| HIM | HER |
| HIS | HER |
| HIMSELF | HERSELF |
| MALE | FEMALE |
| MAN | WOMAN |
| MEN | WOMEN |
| HUSBAND | WIFE |
| FATHER | MOTHER |
| BOYFRIEND | GIRLFRIEND |
| BROTHER | SISTER |
| ACTOR | ACTRESS |

# A   APPENDIX

# B   GENDER-IDENTIFYING WORDS

See Table 2 for the list of gender-identifying words that were masked for prediction during both finetuning and at inference time for the Masked Gender Task, with the exclusion of 'man' & 'woman' that remained unmasked in the extended Winogender evaluation set.

# C   $W$ VARIABLE X-AXIS VALUES

## C.1   PLACE VALUES

Ordered list of bottom 10 and top 10 World Economic Forum Global Gender Gap ranked countries used for the x-axis in Figure 3, that were taken directly without modification from `https://www3.weforum.org/docs/WEF_GGGR_2021.pdf`:

"Afghanistan", "Yemen", "Iraq", "Pakistan", "Syria", "Democratic Republic of Congo", "Iran", "Mali", "Chad", "Saudi Arabia", "Switzerland", "Ireland", "Lithuania", "Rwanda", "Namibia", "Sweden", "New Zealand", "Norway", "Finland", "Iceland"