# OpenReview forum: "Selection Collider Bias in Large Language Models"
_auai.org/UAI/2022/Workshop/CRL — CRL@UAI 2022 Poster_

### Official Review · Reviewer_6sVV · 2022-06-27
**Selection collider bias introduces spurious correlations in LLMs**

**Rating:** 6
**Confidence:** 3

**Review:**

### Summary
This work highlights the problem of selection collider bias in Large Language Models through experimental evaluation. The implicit conditioning on an access variable during the data collection process introduces spurious correlations between unconditionally independent variables that are causing it. To probe how such bias is present in pre-trained LLMs, authors designed a gender pronoun prediction task and analyzed the response curve of models under changes of the independent variables. BERT-like models are evaluated on Wiki-bio and Reddit-TLDR.


### Strengths
* This paper contributes to the new research area of causality in NLP which is of high interest since natural language may be used as general representation modality for data.
* The empirical results of this work shed light on a overlooked problem. It provides experimental evidence on the consequences of collider bias when combined with dataset collection in real-world problems.
* This work challenges the recent scaling trend in LLMs: large language models still suffer from the bias. Therefore, it opens questions about the validity of learnt representation.
* The methodological approach of the experimental prove is elegant and instructive on how we can probe LLMs for causal signals.
* Authors offer insights on how the selection collider bias may be mitigated.


### Weaknesses
* The work could benefit from improvements in terms of presentation, especially considering the broad audience this paper could reach:
	* A rigorous definition of selection bias, collider bias and selection collider bias will help readers understand the problem of interest from the beginning. At the current state,  spurious correlations emerging from the colliders are presented jointly with the selection bias.
	* While the on-going example helps clarifying the concepts, an earlier introduction will be more effective. Authors might consider introducing it from the introduction.
	* The reasoning is IMHO hard to follow. Making it explicit would help understand the presented work: even though considered properties should be independent from gender, as we change them, the prediction changes. Hence, a spurious correlation is injected.
	* The role of the observable `work` is not really clear. Why do the authors introduce it? Can we evaluate the model based on the correctness of prediction for masked words or it is to avoid "predict one’s gender based only on gender-neutral text about them" ?
	* Contributions stated in the introduction paragraph are a bit detached from the rest of the paper. Authors should consider stressing the results of their experiments in the main paper.

* Previous work references do not cover the intersection between language models and causality. Authors might consider extending the literature with recent work on the topic (Egami et al 2018, Jin et al. 2021)

---
Egami, Naoki, et al. "How to make causal inferences using texts." _arXiv preprint arXiv:1802.02163_ (2018).

Jin, Zhijing, et al. "Causal Direction of Data Collection Matters: Implications of Causal and Anticausal Learning for NLP." _EMNLP 2021_. 2021.

---

### Meta-Review · Program_Chairs · 2022-07-06

**Recommendation:** Accept (Poster)
**Confidence:** 2

**Metareview:**

The reviewer made some suggestions on presentation which the authors are encouraged to integrate for a future version, as well as suggesting mentioning recent literature on the intersection between language models and causality. Overall, the paper appears to make a relevant point worth presenting in the workshop.

---

### Decision · Program_Chairs · 2022-07-06

Accept (Poster)